# Set Discrimination Contrastive Learning

## Abstract

In this work, we propose a self-supervised contrastive learning method that integrates the concept of set-based feature learning. The main idea of our method is to randomly construct sets of instances in a mini-batch and then learn to contrast the set representations. Inspired by set-based feature learning, we aggregate set features from individual sample features by a symmetric function. To improve the effectiveness of our set-based contrastive learning, we propose a set construction scheme built upon sample permutation in a mini-batch that allows a sample to appear in multiple sets, which naturally ensures common features among sets by construction, thus, generating hard negative samples. Our set construction scheme also increases both the number of positive and negative sets in a mini-batch, leading to better representation learning. We demonstrate the robustness of our method by seamlessly integrating it into existing contrastive learning methods such as Sim-CLR and MoCo. Extensive experiments demonstrate that our method consistently improves the performance of these contrastive learning methods in various datasets and downstream tasks.

## 1 Introduction

Learning effective representations from data has been a long-standing challenge in machine learning over the past decades. A prominent direction to address this problem is self-supervised learning (SSL), which aims to learn the representations without the need of human supervision. Contrastive learning (Jing & Tian, 2021; Le-Khac et al., 2020) is a modern powerful approach in self-supervised learning that learns a representation based on the idea of attracting and repelling features, *i.e.,* data samples with similar semantics are expected to be close to each other in the feature space while dissimilar samples are expected to stay apart.

A dominant pretext task for contrastive learning is *instance discrimination* where each instance is an original data sample represented by a feature vector. Given an instance, positive samples can be defined as different views of the same instance generated by applying data augmentation to the instance such as cropping and flipping (Ye et al., 2019; Chen et al., 2020a), luminance and chrominance decomposition (Tian et al., 2020a). On the other hand, negative samples are defined as remaining samples such as other samples in the same mini-batch (Ye et al., 2019; Chen et al., 2020a) or they can be generated through memory banks (He et al., 2020; Wu et al., 2018). By distinguishing positive samples from negative samples, effective representations can be learned through such self-supervision. Remarkably, recent progresses demonstrate that self-supervised representations can even surpass the performance of supervised counterparts in computer vision downstream tasks (Henaff, 2020; He et al., 2020).

A limitation of instance discrimination is that this pretext task can be optimized by simply learning low-level features of the data, which might not be effective representations for downstream tasks. This could be due to overfitting when maximizing mutual information between positive views (Tschannen et al., 2020). Task-irrelevant features could also occur when excessive noise is present due to unnecessarily high mutual information between views during learning (Tian et al., 2020b). Unfortunately, it is challenging to identify useful information from noise without any additional cue such as knowledge of downstream tasks.

We conjecture that effective features should be shared among instances *i.e.,* embedding of which should have some degree of mutual information. Our conjecture is based on previous hypothesis that the good bits are those shared between different views of the world (Tian et al., 2020a; Smith &

Gasser, 2005). Specifically, previous works such as Tian et al. (2020a) only take views from the same underlying instance into account while *we expand this conjecture to multiple instances*. Our intuition is that even if instances belong to different categories, they should share some high-level properties such as abstract shapes, part compositions, etc and learning these common concepts might be more beneficial than low-level features. As the same time, instance embedding should be adequately discriminative to be distinguished from each other.

In this paper, we facilitate learning such shared features by considering unordered sets of instances because: 1) the aggregation function in a set-structured representation (Zhang et al., 2020; Naderial-izadeh et al., 2021) devise a bottleneck that encourages the model to learn common features across instances to maximize set mutual information (Section 3.4) ; 2) circumventing instance discrimination helps avoid unintentionally maximizing distances between samples with similar semantics which hinders common features learning.

To realize the idea of set-based learning into self-supervised learning, we propose **M**ultiple **I**nstance **RA**ndomly Grouped for **C**ontrastive **LE**arning, so-called `Miracle`, a simple algorithm for set-based contrastive learning in which we arbitrarily sample data points in a mini-batch and group them to form sets. Similar to instance discrimination (Wu et al., 2018; Ye et al., 2019), we apply data augmentation to create two views of a set. We construct features of a set by aggregating features of the samples of a set by a symmetric function, which can then be passed to a contrastive loss. The network is trained to maximize agreement to views of the same set while being able to distinguish different sets. We refer to this task as *set discrimination*. To support the training, we devise an efficient set construction scheme that is based on permuting the samples in a mini-batch multiple times. The benefits of our set construction is two-fold. First, it allows an instance to appear in multiple sets, and therefore the sets can have common features by construction. This encourages the network to learn common features for the instances, and also generates harder negative sets to improve the robustness of representation learning. Second, our set construction can increase the number of positive and negative sets to improve the self-supervision. In contrast, contemporary methods only focus solely on positives samples (Dwibedi et al., 2021) or negative samples (He et al., 2020; Chen et al., 2020b; Wu et al., 2018) at a time.

By virtue of the simplicity of proposed approach, we can plug this set-based contrastive learning into existing contrastive learning methods. Through extensive experiments, we demonstrate the efficacy of `Miracle` in various scenarios. First, we show that the proposed method consistently improve the baselines such as SimCLR (Chen et al., 2020a), MoCo (He et al., 2020) on CIFAR-10, CIFAR-100, STL-10, ImageNet-100, and ImageNet-1K. We verify the robustness of `Miracle` when scaling up the learning with different hyperparameters including pretraining epochs, batch sizes, learning rates and temperatures. We also study `Miracle` in various conditions including weaker data augmentation and transfer learning.

In summary, our contributions are: (1) a new pretext task of set discrimination for self-supervised visual representation learning; (2) a simple but effective method to integrate set-based feature learning into existing contrastive learning methods, yielding significant performance improvement; (3) extensive experiments and ablation studies that empirically demonstrate the usefulness and robustness of set-based contrastive learning.

## 2 RELATED WORK

**Instance-wise contrastive learning** Recent advances in contrastive learning are largely driven by the *instance discrimination* task (Wu et al., 2018). Prior work (Chen et al., 2020a; He et al., 2020; Ye et al., 2019; Wu et al., 2018; Tian et al., 2020a) in this direction treat each instance as a category and learn an embedding space such that views from same instance, also known as positive samples, obtained by different transformations of an image, should have small distances while views from different instances, or negative samples, should have large distances. There are more effective ways to generate the samples for contrastive learning: Chen et al.; Ye et al. simply use all samples from the same mini-batch; Wu et al. uses a *memory bank* which stores the features from previous steps; He et al. uses a *momentum encoder* to compute positive samples and memory bank for negative samples; Hu et al. trains a generative model together with a representation network to generate negative samples. Most of these methods adopt the InfoNCE loss function (Van den Oord et al., 2018) which usually requires a large batch size to reduce the bias of the estimation. Yeh et al.; Chen et al. propose variants of InfoNCE to cope with aforementioned challenge.

**Cluster-based contrastive learning.** A potential problem of instance-wise contrastive learning is that instance discrimination is performed regardless of the semantic structure of the underlying data, i.e., an instance is attracted to its augmented version and repelled from all other instances despite their similarities. To address this problem, there exists a family of prototypical contrastive learning methods that introduce prototypes of data clusters into their representation learning (Guo et al., 2022; Li et al., 2021a; Caron et al., 2020; Li et al., 2021b; Wang et al., 2021). These methods perform clustering on adapting features during learning, which is a challenging model selection problem (Wang et al., 2021). Our method also considers feature learning on a set of instances, but our grouping is done dynamically for each mini-batch and regardless of instance features.

**Data augmentations.** Data augmentation is a systematic way to boost data diversity in supervised and unsupervised learning with recent applications for self-supervised contrastive learning. A class of data augmentation relevant to our method is image mixtures, e.g., mix-up (Zhang et al., 2018), cutmix (Yun et al., 2019). A few recent works attempted to use image mixtures with contrastive learning such as MixCo (Kim et al., 2020), i-Mix (Lee et al., 2021b), and Un-Mix (Shen et al., 2022). These methods share the same strategy that they first generate mixture samples in the pixel space and then learn to contrast the features among the image mixtures (Shen et al., 2022) or between the image image mixtures with the original views of each image in the mixture (Kim et al., 2020). One can also generate hard negative samples by linear interpolating the embeddings of positive and negatives samples (Kalantidis et al., 2020). Compared to these methods, we *do not* explicitly construct any mixture samples in our learning. We approach the mixture representation from a different perspective by learning to contrast set features aggregated on-the-fly during training.

**Set-based learning.** Learning features from a set of data points is essential and has been explored in machine learning and computer vision under the umbrella of multiple instance learning (Dietterich et al., 1997; Ilse et al., 2018), 3D shape recognition (Qi et al., 2017a;b). A crucial requirement for such set representation is *permutation invariance i.e.,* the output of the model is invariant to the order of each input instance. Existing works achieve this condition via a symmetric function, commonly implemented by pooling layers *e.g.,* max, mean, sum, etc. to aggregate features from all instances of a given set. More attempts have been made to devise more sophisticate and learnable aggregation functions (Skianis et al., 2020; Mialon et al., 2020; Murphy et al., 2019; Lee et al., 2019; Zhang et al., 2020; Naderializadeh et al., 2021). Here our method integrates the concept of feature aggregation and permutation invariance in set-based learning to the self-supervised contrastive learning framework.

# 3 METHODOLOGY

In this section, let us first recall the preliminary of contrastive learning (Section 3.1), and then introduce our proposed set-based contrastive learning (Section 3.2) and its implementation details (Section 3.3). We then provide a discussion on why and how our set discrimination works from different perspectives (Section 3.4).

## 3.1 CONTRASTIVE LEARNING

Given a set $\mathbf{X} = \{\mathbf{x}_1, \mathbf{x}_2, \cdots, \mathbf{x}_N\}$ of $N$ of unlabeled data samples, we aim to learn a function $f$ that maps $\mathbf{x}_i$ to a low-dimensional embedding $\mathbf{h}_i$ on an unit hypersphere *i.e.,* $f \in \mathcal{F} : \mathbb{R}^d \to \mathbb{S}^m$. This mapping can be learned with *instance discrimination* in contrastive learning. Specifically, in a mini-batch, a positive sample pair is generated by applying data augmentation $t, t' \sim \mathcal{T}$ to a sample $\mathbf{x}_i$, and pairs that involve augmentations of remaining samples are negative samples. The network $f$ can be trained to maximize the agreement between two augmented views of the same instance, and minimize that of views from different instances by using InfoNCE (Van den Oord et al., 2018) loss:

$$\mathcal{L}_{\text{InfoNCE}} = -\log \frac{\exp(\text{sim}(\mathbf{z}_i, \mathbf{z}_j)/\tau)}{\sum_{k=1}^{2N} \mathbb{1}_{[k \neq i]} \exp(\text{sim}(\mathbf{z}_i, \mathbf{z}_k)/\tau)}, \tag{1}$$

where $\mathbf{z}_i = g(\mathbf{h}_i)$ and $\mathbf{z}_j = g(\mathbf{h}_j)$ are projected vectors of the embedding $\mathbf{h}$ given a projection function $g$; $\tau$ is the temperature to control the confidence of the feature similarity between two vectors using $\text{sim}(u, v) = u^\top v / (\|u\| \|v\|)$.

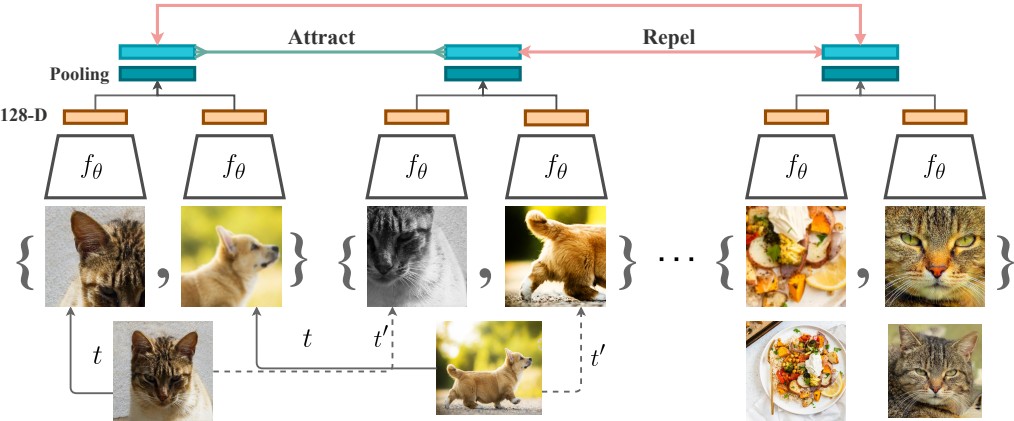

Figure 1: Training pipeline of our proposed method. Similar to conventional *instance discrimination* task, we apply random augmentation to each image to obtain two views of corresponding instance. Then, we **randomly** pair $K$ instances to construct a set and apply permutation-invariant pooling on the projected embedding to obtain set representation $\in \mathbb{S}^{127}$. We simply plug this representation for existing contrastive learning methods such as SimCLR (Chen et al., 2020a), MoCo (He et al., 2020).

## 3.2 SET-BASED CONTRASTIVE LEARNING

We take this idea of *instance discrimination* (Wu et al., 2018; Ye et al., 2019) a step further by generalizing the representation learning to a set of samples, letting the network learn features invariant to a set. An overview of our approach is depicted in Figure 1.

Let us briefly recall the basic idea of set-based learning. To construct the model for processing set-structured data, we can leverage a *set pooling model* (Zaheer et al., 2017). In this model, each instance in a set is first fed into a neural network independently. Then, we use an aggregation function $\phi$ to obtain the global signature of a set where $\phi$ is a symmetric function, which allows its output to be invariant to the order of the input elements. More formally:

$$\text{net}(\{\mathbf{x}_1, \mathbf{x}_2, \cdots, \mathbf{x}_n\}) = g(\phi(f(\mathbf{x}_1), f(\mathbf{x}_2), \cdots, f(\mathbf{x}_n))) \qquad (2)$$

where $f, g$ are neural networks, $\phi$ is an aggregation function that can be implemented by a pooling layer. Zaheer et al. (2017) proved that the above composition provides a universal approximator for any permutation-invariant functions. This set-based learning is also widely adopted for point cloud feature learning in 3D computer vision (Qi et al., 2017a;b).

We propose to apply the idea of set-based learning to self-supervised contrastive learning as follows. Given a mini-batch of size $B$, we construct a set of $K$ random samples $\{\mathbf{x}_1, \cdots, \mathbf{x}_K\}$ from the mini-batch. We then apply data augmentations $t, t' \sim \mathcal{T}$ to generate two sets of $K$ augmented samples $\{\mathbf{y}_1, \cdots, \mathbf{y}_K\}$ and $\{\mathbf{y}'_1, \cdots, \mathbf{y}'_K\}$, where $\mathbf{y}_k = t(\mathbf{x}_k)$ and $\mathbf{y}'_k = t'(\mathbf{x}_k)$ for $k \in \{1, ..., K\}$. The representation of the sets can be written as:

$$\mathbf{h} = \phi\left(g(f(\mathbf{y}_1)), \cdots, g(f(\mathbf{y}_K))\right), \qquad (3)$$
$$\mathbf{h}' = \phi\left(g(f(\mathbf{y}'_1)), \cdots, g(f(\mathbf{y}'_K))\right),$$

where we followed SimCLR (Chen et al., 2020a) and use an encoder $f$ and projection head $g$ to obtain the features of the sample in each set. Note that in our formulation we place the aggregation function $\phi$ at last. In fact, we can add another projection head after $\phi$ as done in the architecture by Zaheer et al. (2017). However, we empirically found that the difference in performance with and without the additional projection head is negligible. Here we use the formulation with the aggregation at the end and report variants of this design in the appendix.

## 3.3 INPUT PERMUTATION

Given a mini-batch of $B$ samples, we need to generate sets of size $K$ for our set-based contrastive learning. A naive approach is to randomly group every $K$ samples in the batch to construct a set.

However, this yields inferior performance compared to conventional instance discrimination. This is because this approach can only construct a limited number of sets with $\lceil \frac{B}{K} \rceil$ positive pairs and $2(\lceil \frac{B}{K} \rceil - 1)$ negative samples for each mini-batch. The utilization of the samples in a batch is not optimal because each sample is only considered once during set construction.

To address this problem, we propose to permute input samples multiple times to augment **both** positive and negative samples for each training step. Particularly, we first fed the $B$ input samples to the feature extractor (before the pooling layer). Next, we create a permutation matrix $\pi$ of size $M \times B$ where each row of the matrix is a permutation of indexes ranging from 0 to $B$ and $M$ is the number of times we shuffle the input. Each set in the mini-batch can then be constructed by gathering corresponding $K$ elements from the permutation matrix. Finally, we perform pooling and (optionally) fed those aggregated vectors to the last MLP.

This approach effectively increases the number of positive and negative pairs and extends a mini-batch for a total of $BM/K$ sets. Another benefit of this set construction scheme is that it allows an instance to appear in multiple sets, which means that some sets can naturally share some common features by construction. When these sets form negative pairs, these pairs can be regarded as hard negatives which can improve the robustness of the representation learning. We empirically demonstrate the importance of our set construction step in Section 4.2 and Figure 2.

---

**Algorithm 1:** Pseudocode of our method in SimCLR style.

```
# batch B, permutation size M, set size K
# encoder f, projection g, augmentation
 t,t' ~ 𝒯
for sampled minibatch {x_k}^B_{k=1} :
   for k ∈ {1,...,B} :
      y_k = t(x_k),  y'_k = t'(x_k)
   let π = permutation matrix of size M × B
   let N = BM/K, let z = []
   for k ∈ {1,...,M} :
      for i ∈ {1,...,B/K} :
         ψ = π[k,(i−1)K + 1 : iK]
         z = φ(g(f(y_{ψ(1)})),⋯,g(f(y_{ψ(K)})))
         z' = φ(g(f(y'_{ψ(1)})),⋯,g(f(y'_{ψ(K)})))
      z.append(z,  z')
   for i ∈ {1,...,2N} and j ∈ {1,...,2N} :
      s_{i,j} = z_i^⊤ z_j/(‖z_i‖‖z_j‖)
   let ℓ(i,j) = − log ( exp(s_{i,j}/τ) / Σ^{2N}_{k=1} 1_{k≠i} exp(s_{i,j}/τ) )
   𝓛 = (1/2N) Σ^N_{k=1} ℓ(2k−1,2k) + ℓ(2k,2k−1)
   minimize 𝓛 and update f and g
return f and discard g
```

---

### 3.4 DISCUSSION

**Set-based contrastive learning encourages common features.** We explain why set discrimination prioritize common features than low-level features from the instances with the following theorem.

**Theorem 1.** *Let $X_1, X_2$ be two instances of a set and $Z, Z'$ be the set representation from two views of previous instances. The objective function of set discrimination is the lower bound of:*

$$\text{InfoNCE}(Z; Z') \leq I(Z; X_1) + I(Z; X_2) + I(X_1; X_2 | Z)$$

The proof can be found in the appendix for interested readers. We can observe that the upper bound of objective function in set discrimination shares similarities with the objective in Deep InfoMax (Hjelm et al., 2019). Particularly, the "global" context, *i.e.,* set representation $Z$ is encouraged to have high mutual information with all the instances of that set. Moreover, the set representation is bottlenecked because its vector has the same dimension as the vector of each instance representation. If the encoder learns features that are specific to only one instance of the set, it does not increase the mutual information with any of the other instances of that set. Therefore, this training scheme favors encoding aspects of the data that are shared across elements.

We also note that attracting or repelling the aggregated representation in contrastive learning directly leads to optimizing the representation of each individual sample in the set. This is an important factor that separates our method from cluster-based contrastive learning, where the cluster prototypes act as a regularization in additional to the learning of *instance discrimination* pretext task. By contrast, we find that simply training the model with only *set discrimination* task without using any instance discrimination loss can already lead to performance gain.

**Set discrimination from hard negatives perspective.** To discriminate between sets, a set representation that can preserve more information from all instances is desirable. This is because given that

our set construction scheme ensures an instance to appear in multiple sets, if the output only favors representative features of an individual sample in a set, the network will eventually produce identical representations for multiple sets, rendering it even harder to distinguish between these sets. From the perspective of negative sample mining, our set construction can also be regarded as a strategy to produce hard negatives.

**Set discrimination from data augmentation perspective.** Compare to SDMP (Liu et al., 2022), i-mix (Lee et al., 2021b), mix-Co (Kim et al., 2020), `Miracle` can be viewed as a more general and principled way for data augmentation. For instance, i-mix mixes the input images together in image space, which can be challenging to adopt to two views from different modalities e.g., image and language. On the contrary, our method operates on latent space and has the potential to be applicable for such mixing. There are multiple options to select the aggregation function $\phi$ as a symmetric function: average pooling, max pooling, Wasserstein pooling (Naderializadeh et al., 2021), attention pooling (Lee et al., 2019), etc. More results are provided in Table 5 in Section 4.2.

## 4    EXPERIMENTS

We conduct experiments and empirically verify the effectiveness of proposed method `Miracle` when plugging it to different contrastive learning baselines. More experiments on transfer learning, etc can be found in the appendix. Note that for simplicity, we do not use multi-crop augmentation.

### 4.1    EXPERIMENTAL SETUP

We conduct experiments on both small-scaled benchmarks: CIFAR-10/CIFAR-100 (Krizhevsky et al., 2009), STL-10 (Coates et al., 2011) and large-scaled benchmarks: ImageNet-1K (Deng et al., 2009), ImageNet-100 (Tian et al., 2020a).

**Small-scaled datasets.** For CIFAR-10, CIFAR-100, and STL-10, we use ResNet-18 (He et al., 2016) as the backbone model and remove the MaxPool layer of it according to conventional. We follow the hyper-parameters of of CLD (Wang et al., 2021). The configuration in this setting is similar to ImageNet setup except the temperature $\tau$ is set to $0.07$, and we use weight decay of $0.0005$ for all experiments. The data augmentation strategies are adopted from CLD and provided in the appendix.

**Large-scaled datasets.** For ImageNet-1K and ImageNet-100, we implement our method following SimCLR (Chen et al., 2020a). Specifically, we use ResNet-50 (He et al., 2016) as the backbone network. We use SGD optimizer with learning rate $\eta = 0.03 \times \mathrm{batch\_size}/256$, weight decay of $0.0001$, momentum of $0.9$. We utilize the cosine annealing learning rate schedule and warm-up the learning rate for the first 5 epochs by linearly increase it to $\eta$. We set the temperature $\tau$ to $0.1$ and latent vector has the dimension of $128$.

**Evaluation.** For *linear probing* benchmark, we adopt standard setup as in (Li et al., 2021a; He et al., 2020). Specifically, we train the linear layer for $100$ epochs with SGD optimizer with the learning rate of $30$, weight decay of $0$, and momentum of $0.9$. The learning rate is decayed by a factor of $0.1$ at epoch $60$ and $80$, batch size is set to $256$. For k-nearest neighbors (kNN) evaluation, we set the number of neighbors to $200$.

**`Miracle` configuration.** We set number of elements per set to $K = 2$, because it is sufficient for learning common features as in Theorem 1. We use the global average pooling (GAP) as our aggregation function. We experiment with permutation size $M = 32$ unless otherwise stated. For evaluation, we extract the same features as in SimCLR (Chen et al., 2020a).

### 4.2    MAIN RESULTS

We demonstrate the effectiveness of `Miracle` while incoporating it to other self-supervised methods: SimCLR (Chen et al., 2020a), MoCo (He et al., 2020). We use the same training and evaluation setup between all approaches for fair comparison.

**CIFAR and STL.** We first evaluate the performance of `Miracle` with different batch sizes and compare it to the baseline SimCLR (Chen et al., 2020a) and MoCo (He et al., 2020). From Table 1, we can observe that integrating our framework consistently increases the performance for all scenarios. It is worth noticing that for STL-10, we train all self-supervised models on the *"train+unlabeled"* split similar to Wang et al. (2021).

Table 1: Comparison of `Miracle` with SimCLR and MoCo baselines on CIFAR-10, CIFAR-100, STL-10 with **k-nearest neighbors**. The 95% confidence interval with 3 runs is approximately 0.3 for all experiments.

| | CIFAR-10 | | | | CIFAR-100 | | | | STL-10 | | | |
|---|---|---|---|---|---|---|---|---|---|---|---|---|
| Batch size | 128 | 256 | 512 | 1024 | 128 | 256 | 512 | 1024 | 128 | 256 | 512 | 1024 |
| SimCLR | 80.1 | 81.2 | 81.7 | 81.6 | 51.6 | 52.3 | 52.7 | 53.1 | 76.7 | 77.6 | 76.6 | 76.3 |
| w/ `Miracle` | **85.8** | **86.2** | **86.2** | **86.1** | **55.4** | **56.6** | **57.1** | **57.1** | **84.7** | **85.2** | **84.6** | **84.5** |
| MoCo-v2 | 82.1 | 82.6 | 81.9 | 81.2 | 53.2 | 53.2 | 53.8 | 53.0 | 79.8 | 79.0 | 79.3 | 79.3 |
| w/ `Miracle` | **86.1** | **86.6** | **85.8** | **85.5** | **56.3** | **58.1** | **57.3** | **57.1** | **84.7** | **85.0** | **84.0** | **84.1** |
| DCL | 83.1 | 82.7 | 82.3 | 82.1 | 52.7 | 53.3 | 53.5 | 53.1 | 81.6 | 81.1 | 81.2 | 81.1 |
| w/ `Miracle` | **86.1** | **86.7** | **86.6** | **86.6** | **56.3** | **57.6** | **57.7** | **57.7** | **84.9** | **85.3** | **85.8** | **85.6** |
| FlatNCE | 82.9 | 83.0 | 82.5 | 82.4 | 53.7 | 53.8 | 53.2 | 53.5 | 81.5 | 81.0 | 81.2 | 80.9 |
| w/ `Miracle` | **86.5** | **86.5** | **86.6** | **86.3** | **56.4** | **57.1** | **57.5** | **57.2** | **84.6** | **85.8** | **85.4** | **85.1** |

Table 2: Comparison between our `Miracle` and SimCLR (Chen et al., 2020a) on ImageNet-100 with *k-nearest neighbors* and *linear probing* evaluation.

| | kNN | | | | | | Linear probing | | | | | |
|---|---|---|---|---|---|---|---|---|---|---|---|---|
| | Top-1 | | | Top-5 | | | Top-1 | | | Top-5 | | |
| Batch size | 256 | 512 | 1024 | 256 | 512 | 1024 | 256 | 512 | 1024 | 256 | 512 | 1024 |
| SimCLR | 55.6 | 56.0 | 57.2 | 83.9 | 84.0 | 84.7 | 70.6 | 69.5 | 70.7 | 90.2 | 90.5 | 91.0 |
| w/ `Miracle` | **65.6** | **67.3** | **67.8** | **90.2** | **90.5** | **90.4** | **77.2** | **77.0** | **76.1** | **93.8** | **93.6** | **93.0** |

We also conduct experiments to examine whether our proposed method can give benefits to these variants, namely, Decoupled Contrastive Learning (Yeh et al., 2021), FlatNCE (Chen et al., 2021a). Table 1 reports the performance of our proposed approach with aforesaid objective functions. We can observe that these loss functions indeed boost the performance of contrastive models on all examined datasets. At the same time, integrating `Miracle` with these losses also leads to superior performances compared to vanilla SimCLR and slightly better than using `Miracle` with InfoNCE.

**ImageNet.** For ImageNet-100, in Table 2, we report the top-1 and top-5 accuracy with linear evaluation of `Miracle` and SimCLR baselines. For ImageNet-1K, we report the top-1 accuracy evaluated with linear probing of `Miracle` in Table 3. We denote our implementation for the baseline with conventional SGD optimizer and the aforesaid configuration as SimCLR, respectively. In addition, we also study the efficacy of with more optimal/well-tuned hyper-parameters for SimCLR as suggested in (Chen et al., 2020a; 2021b), which are indicated as SimCLR+. More concretely, ResNet-50 network and 4-layers MLP projection head is trained with LARS optimizer (You et al., 2017) and square learning rate scaling.

## 4.3 WEAKER DATA AUGMENTATION

As aforementioned, another perspective to view our approach is through the lens of *data augmentation*. Therefore, we compare our method with SimCLR (Chen et al., 2020a) where we using different groups of data augmentation techniques. Here we report the performance of SimCLR (Chen et al., 2020a) and our method when a specific data augmentation technique is removed from both methods. As can be seen in Table 4, with fewer data augmentation techniques, the accuracy of both SimCLR and our method is decreased, but our method can still outperform SimCLR by a wide margin, proving the effectiveness of our set-based contrastive learning. This also proves that our method is orthogonal to existing data augmentation techniques used by SimCLR (Chen et al., 2020a).

## 4.4 ABLATION STUDY

In this section, we conduct a wide range of ablation studies to further articulate and analyze the behavior of `Miracle` as well as its robustness on various scenarios.

**Choice of aggregation functions.** For simplicity, in all previous experiments, we use global average pooling (GAP) as the aggregate function $\phi(\cdot)$. Here we adopt other pooling implementations and

Table 3: Comparison between our `Miracle` and Sim-CLR (Chen et al., 2020a) on ImageNet-1K with *linear probing* evaluation.

Table 4: Top-1 kNN accuracy of SimCLR and `Miracle` on CIFAR-10 and CIFAR-100 without a specified data augmentation. **GB** denotes *Gaussian blurring*, **CJ** denotes *color jittering*, **HF** denotes *horizontal flipping*, **RC** denotes *random cropping*. In each run, we remove **only** the specified augmentation from training.

| Batch size | ImageNet-1K | |
|---|---|---|
| | 256 | 1024 |
| SimCLR | 61.5 | 68.0 |
| w/ `Miracle` | **65.2** | **68.6** |

| without | CIFAR-10 | | | | CIFAR-100 | | | |
|---|---|---|---|---|---|---|---|---|
| | **GB** | **CJ** | **HF** | **RC** | **GB** | **CJ** | **HF** | **RC** |
| SimCLR | 80.3 | 71.9 | 80.1 | 40.6 | 51.8 | 41.8 | 50.5 | 15.5 |
| `Miracle` | **85.9** | **79.0** | **85.1** | **49.1** | **56.9** | **43.8** | **54.4** | **22.3** |

Table 5: Top-1 accuracy of `Miracle` with various aggregation functions on CIFAR-10, CIFAR-100, STL-10, and ImageNet-100. All methods are pretrained for 200 epochs with batch size of 256. ResNet-18 is used as the backbone network for CIFAR/STL while we adopt ResNet-50 for ImageNet-100. For evaluation, *k-nearest neighbor* is used for CIFAR-10, CIFAR-100, STL-10 and *linear probing* is used for ImageNet-100.

| Aggregation | CIFAR-10 | CIFAR-100 | STL-10 | ImageNet-100 |
|---|---|---|---|---|
| SimCLR | 81.2 | 52.3 | 77.6 | 70.6 |
| GAP | 86.2 | **56.6** | **85.2** | **77.2** |
| MaxPool | **86.3** | 56.1 | 84.6 | **77.2** |
| PMA (Lee et al., 2019) | 85.7 | 55.9 | 84.4 | 76.9 |
| FSPool (Zhang et al., 2020) | 85.7 | 55.5 | 84.4 | 76.5 |
| PSWE (Naderializadeh et al., 2021) | 85.4 | 55.1 | 84.2 | 76.9 |

compare their performance. We conduct experiments with MaxPool, Featurewise Sort Pooling (Zhang et al., 2020), Pooling with Sliced-Wasserstein Embedding (PSWE) (Naderializadeh et al., 2021), and Pooling by Multihead Attention (PMA) (Lee et al., 2019). We report the performance of `Miracle`-32 with aforementioned aggregation methods in Table 5. Perhaps surprisingly, GAP achieves the highest accuracy compared to other sophisticated algorithms. Nevertheless, we can observe that our proposed approach consistently outperforms the vanila contrastive learning on all datasets regardless of the choice of the aggregation function. These empirical results demonstrate the robustness of `Miracle` with a wide range of aggregation functions.

**Number of input permutation** We analyze the importance of input permutation to increase the number of positive and negative sets in Figure 2. We denote `Miracle`-m as proposed method with $M = m$. Without input permutation, *i.e.,* `Miracle`-1, set discrimination results in inferior performance than instance discrimination. However, with permutation of size $M = 2$, which implies the same number of positive and negative samples in a mini-batch for both `Miracle` and SimCLR (Chen et al., 2020a), our method can immediately outperform the baseline. Using $M > 2$ leads to even better performance.

**Number of elements of a set.** By default we only construct sets of size $K = 2$. We now extend the number of instances per set (*i.e.,* set size) to a larger number and examine the performance of learned models. Intuitively, increasing the number of elements per set can lead to more challenging optimization problems since learning common features of the instances in the set becomes more difficult, and therefore the network cannot preserve as much information in a large set compared to a smaller one. Figure 3 shows the top-1 accuracy of models with different set sizes. As the number of possible sets is $BM/K$ (Section 3.4), increasing $K$ reduces significantly the number of positive and negative sets, which deteriorates the performance. However, with enough input permutation, we found that the value of set size does not have a strong effect on the final accuracy.

**Hyperparameters sensitivity.** We investigate the sensitivity of our approach to hyperparameters such as temperature $\tau$ and learning rate $\eta$ compared to SimCLR on CIFAR-100 and STL-10. Figure 3c and 3d report the top-1 accuracy of `Miracle` compared to SimCLR when varying the temperature $\tau$. We observe that `Miracle` significantly and consistently outperforms SimCLR for most temperature values. Furthermore, `Miracle` seems to be more robust to temperature changing than SimCLR. Figure 4 reports the top-1 accuracy of `Miracle` compared to SimCLR when varying the learning rate $\eta$. It shows that `Miracle` consistently outperforms SimCLR for all learning rate values.

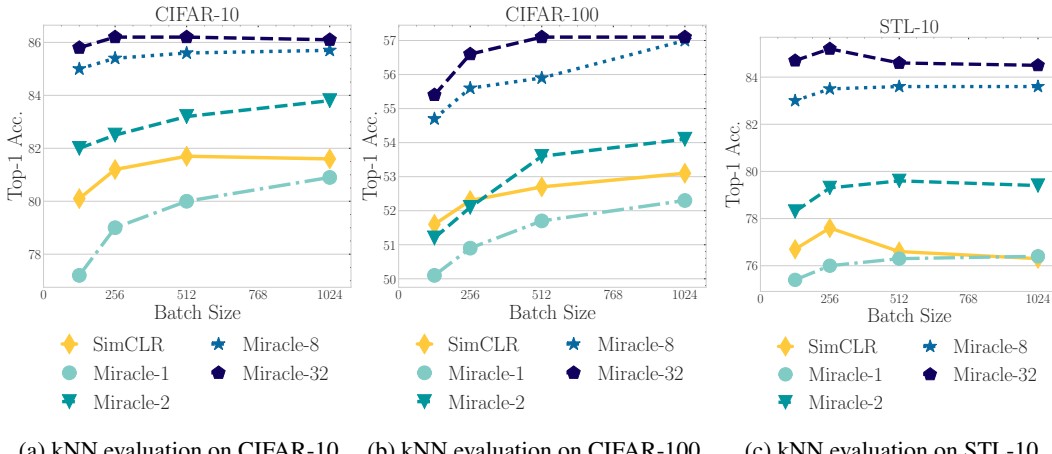

(a) kNN evaluation on CIFAR-10 (b) kNN evaluation on CIFAR-100 (c) kNN evaluation on STL-10

Figure 2: Performance of `Miracle` with different input permutation of size $M \in \{1, 2, 8, 32\}$ on CIFAR-10/100, and STL-10 with varying batch sizes. With the same number of positive and negative samples per batch *i.e.,* `Miracle`-2, our method already achieves better performance than SimCLR. Additional accuracy gain can be obtained by increasing number of positive and negatives via $M$.

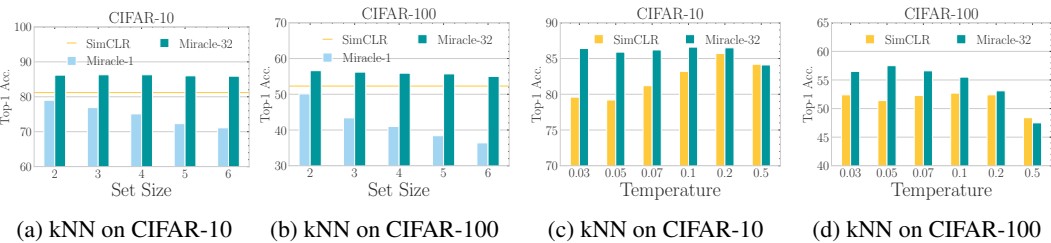

(a) kNN on CIFAR-10 (b) kNN on CIFAR-100 (c) kNN on CIFAR-10 (d) kNN on CIFAR-100

Figure 3: Performance of `Miracle`-32 and SimCLR with different set sizes and temperature. All models are trained with batch size of 256. We use ResNet-18 and k-nearest neighbors for evaluation.

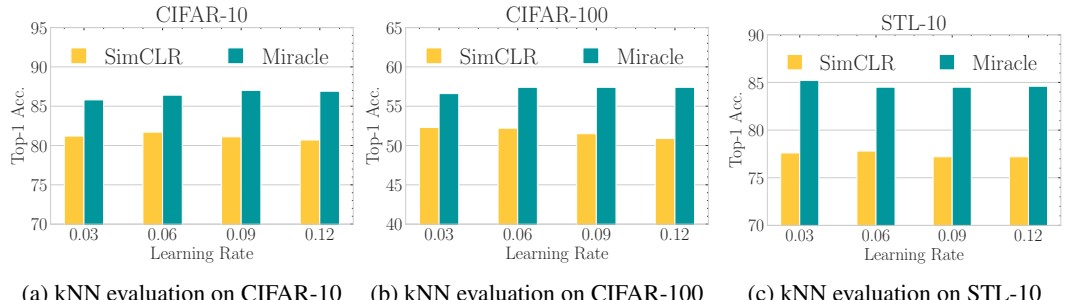

(a) kNN evaluation on CIFAR-10 (b) kNN evaluation on CIFAR-100 (c) kNN evaluation on STL-10

Figure 4: Performance of `Miracle` and SimCLR when varying learning rate value on CIFAR-10, CIFAR-100, and STL-10. All models are pretrained for 200 epochs with batch size of 256 and ResNet-18 as backbone. `Miracle` consistently outperform SimCLR for all learning rate values.

## 5 CONCLUSION

In this paper, we introduce the novel idea of incorporating set-structured learning to contrastive learning, resulting in the set discrimination pretext task. Our method brings a simple but effective extension to existing contrastive learning methods including SimCLR and MoCo, encouraging the learning of instance features via contrasting positive and negative set features. Our experiments demonstrate the effectiveness and robustness of our proposed method on various scenarios. Our finding is potential in that it is orthogonal to existing data augmentations and image mixtures used in contrastive learning, and hence opens the opportunities to further increase the performance by combining all these methods together.

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

In this appendix, we provide additional experiments of our method in various scenarios. First, we provide proof for our theorem in main paper (Section A). We show our method is orthogonal to existing data augmentations in SimCLR. Even though with fewer augmentation techniques, our method can still improve upon SimCLR (Section 4.3). We also compare our method in the scenario of transfer learning (Section B). We then report more ablation studies (Section C), and the implementation details (Section D). For clarity, we include all table details for the plots in the main paper as well (Section E).

## A  PROOF

*Theorem 1*. Consider two random variables represent first and second element of a set: $X_1, X_2$. We denote extracted features from view 1 of $X_1$ and $X_2$ as $Z_1$ and $Z_2$, respectively. Similarly, $Z_1', Z_2'$ are extracted features from view 2. Let $Z$ and $Z'$ be aggregated features for set representation from view 1 and 2, respectively. The probabilistic graphical model can be illustrated as: $Z \leftarrow (Z_1, Z_2) \leftarrow (X_1, X_2) \rightarrow (Z_1', Z_2') \rightarrow Z'$. Above model is I-equivalent to $Z \rightarrow (Z_1, Z_2) \rightarrow (X_1, X_2) \rightarrow (Z_1', Z_2') \rightarrow Z'$. Applying the *data processing inequality* to above Markov chain, the objective in set discrimination has an upper-bound of:

$$\text{InfoNCE}(Z; Z') \leq I(Z'; Z) \leq I(Z; (X_1, X_2)) = H(X_1, X_2) - H(X_1, X_2 | Z) \quad (4)$$

Since we randomly pair $X_1$ and $X_2$, they are independent to each other, *i.e.*, $H(X_1, X_2) = H(X_1) + H(X_2)$. In addition, we have $H(X_1, X_2 | Z) = H(X_1 | Z) + H(X_2 | Z) - I(X_1; X_2 | Z)$.

Replacing above equations to Equation 4, we obtain:

$$
\begin{aligned}
I(Z; (X_1, X_2)) &= H(X_1, X_2) - H(X_1, X_2 | Z) \\
&= H(X_1) + H(X_2) - \Big[ H(X_1 | Z) + H(X_2 | Z) - I(X_1; X_2 | Z) \Big] \\
&= \Big[ H(X_1) - H(X_1 | Z) \Big] + \Big[ H(X_2) - H(X_2 | Z) \Big] + I(X_1; X_2 | Z) \\
&= I(Z; X_1) + I(Z; X_2) + I(X_1; X_2 | Z) \quad \square
\end{aligned}
$$

## B  TRANSFER LEARNING

We analyze the benefit of `Miracle` for transfer learning on 10 image classification datasets including: CIFAR-10/100, STL-10, Food, Pets, StanfordCars, Oxford Flowers, Oxford Pets, Caltech-101, CUB-200. We use the linear protocol as in Chen et al. (2020a); Lee et al. (2021a). Particularly, we train a linear classifier on extracted features from freezed pretrained models. For both training and testing, images are resized to 224 pixels along the shorter size with bicubic interpolation. We then center crop input images by $224 \times 224$. The linear classifier is trained with L-BFGS optimizer and $\ell_2$-regularized cross-entropy loss. The regularization hyper-parameter is chosen from 45 logarithmically spaced values ranged from $10^{-6}$ to $10^5$. After validation, we train the classifier again with both *train* and *val* data. Note that our train/test split follow Lee et al. (2021a).

Table 6 shows the result of transfer learning with Resnet-50 pretrained on Imagenet-1K with batch size of 256 for 200 epochs. More concretely, aforesaid SimCLR and our `Miracle` achieve top-1 accuracy of 61.5 and 65.2 with linear probing on Imagenet-1K, respectively. We report the top-1 accuracy of all datasets except those marked with *, which we adopt class average accuracy similar to Lee et al. (2021a).

## C  ADDITIONAL ABLATION STUDIES

**Use of additional projection head.**  For simplicity, we do not use any layers after the aggregation functions $\phi$ in all previous experiments. In this section, we conduct an ablation study to investigate the effect of different choices of additional layers after the aggregation function $\phi$. We denote this additional function as $h(\cdot)$. Thus, the representation of a set of size $n$ consisting of $\{\mathbf{x}_1, \cdots, \mathbf{x}_n\}$ is given by $h(\phi(g(f(\mathbf{x}_1)), \cdots, g(f(\mathbf{x}_n))))$. Table 7 reports the performance of `Miracle` with

Table 6: Performance of SimCLR Chen et al. (2020a) and our `Miracle` pretrained on Imagenet-1K for 200 epochs with batch size of 256 when transfering to other datasets. We use linear probing with LBFGS optimizer for all experiments. * denotes datasets evaluated with *class average accuracy*, other datasets are evaluated with top-1 accuracy.

| Dataset | CIFAR10 | CIFAR100 | STL10 | Food | Cars | CUB | Flowers* | Pets* | Caltech101* |
|---|---|---|---|---|---|---|---|---|---|
| SimCLR | 81.6 | 60.1 | 92.1 | **65.2** | 36.1 | 31.5 | **86.1** | 70.5 | 84.1 |
| Miracle | **86.7** | **63.7** | **95.0** | 64.4 | **36.9** | **33.4** | 85.4 | **76.0** | **86.0** |

different choices of $h(\cdot)$. We found that using $h$ as an identity function yields the best results. Other functions such as linear or MLP does not yield improvement.

Table 7: Performance of `Miracle`-32 with different setups of of layers after aggregation *i.e.,* $h(\cdot)$ on CIFAR-10, CIFAR-100, STL-10 evaluated with **k-nearest neighbors**.

| Batch size | CIFAR-10 | | | | CIFAR-100 | | | | STL-10 | | | |
|---|---|---|---|---|---|---|---|---|---|---|---|---|
| | 128 | 256 | 512 | 1024 | 128 | 256 | 512 | 1024 | 128 | 256 | 512 | 1024 |
| SimCLR | 80.1 | 81.2 | 81.7 | 81.6 | 51.6 | 52.3 | 52.7 | 53.1 | 76.7 | 77.6 | 76.6 | 76.3 |
| $h$ = Identity | 85.8 | 86.2 | 86.2 | 86.1 | 55.4 | **56.6** | **57.1** | **57.1** | **84.7** | **85.2** | **84.6** | **84.5** |
| $h$ = Linear | 85.7 | 85.8 | **86.2** | 86.0 | **55.5** | 56.3 | 56.6 | 56.7 | 84.5 | 84.8 | - | - |
| $h$ = MLP | 85.6 | 85.9 | 85.8 | 85.8 | 55.0 | 55.6 | 56.5 | 56.5 | 84.4 | 84.3 | - | - |

**Comparisons with other methods.**  We compare `Miracle` with other self-supervised learning algorithms such as SimSiam Chen & He (2021), BYOL Grill et al. (2020), etc. on small-scale datasets in Table 9. In these experiments, we pretrain `Miracle` for 200 epochs with batch size of 256 and learning rate of 0.06. We train the linear classifier for 100 epochs with batch size of 256 and SGD optimizer. More precisely, we set the learning rate to 30, momentum to 0.9 and weight decay to 0. As can be seen, our method outperforms all other methods on CIFAR-100, STL-10, and ImageNet-100, and works comparably (0.4% difference) with SimSiam Chen & He (2021) on CIFAR-10.

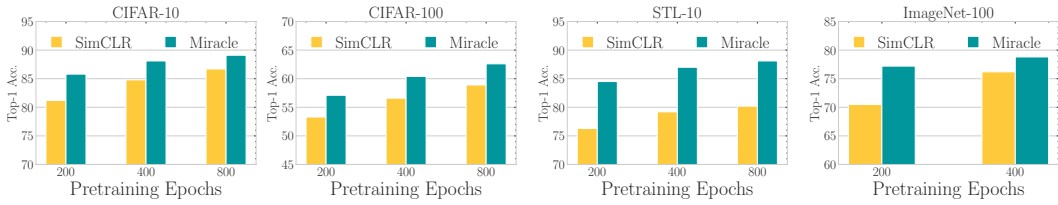

Figure 5: Performance of `Miracle`-32 and SimCLR when scaling up the pre-training budget. All models are trained with batch size of 1024. We use ResNet-18 and k-nearest neighbors for evaluation on CIFAR, STL, and use ResNet-50 and linear probing for evaluation on ImageNet-100.

**Performance with more training budgets.**  It is well established that contrastive learning benefits more from both longer training and larger batch size than its supervised counterpart. Here, we investigate whether our proposed method remains effective when expanding the scale of the pretraining phase. We report the performance of SimCLR (Chen et al., 2020a) and `Miracle` in Figure 5. We observe that `Miracle` remains effective when scaling up the pretraining budget. For small-scale datasets, the performance gap between `Miracle` and SimCLR (Chen et al., 2020a) remains large when scaling up the number of epochs. For large-scale datasets like ImageNet-100, the gap between `Miracle` and SimCLR (Chen et al., 2020a) becomes smaller at 400 epochs but `Miracle` remains more accurate than SimCLR (Chen et al., 2020a).

Table 8: Top-1 accuracy comparison on linear evaluation between `Miracle` and existing methods on ImageNet-1K. All models are pretrained for 200 epochs with ResNet-50 backbone. Cells, which report our implementation, are shaded in green. ME and MB are Momentum Encoder and Memory Bank, respectively. * denotes number taken from (Xu et al., 2021). SwAV (Caron et al., 2020) uses multi-crop to booster the performance of trained network.

| Method | Batch | ME | MB | T1 Acc. | Method | Batch | ME | MB | T1 Acc. |
|---|---|---|---|---|---|---|---|---|---|
| NPID (Wu et al., 2018) | 256 | ✗ | ✓ | 58.5 | PIRL (Misra & Maaten, 2020) | 1024 | ✗ | ✓ | 63.6 |
| MoCo v2 (He et al., 2020) | 256 | ✓ | ✓ | 67.5 | CPCv2 (Henaff, 2020) | 512 | ✗ | ✗ | 67.6 |
| PCL (Li et al., 2021a) | 256 | ✓ | ✓ | 67.6 | MoCHi (Kalantidis et al., 2020) | 512 | ✓ | ✓ | 67.6 |
| CPC v2 (Henaff, 2020) | 256 | ✗ | ✗ | 63.8 | PIC (Cao et al., 2020) | 512 | ✗ | ✓ | 67.6 |
| SimSiam (Chen & He, 2021) | 512 | ✗ | ✗ | **70.0** | MoCo v2* (He et al., 2020) | 1024 | ✓ | ✓ | 67.5 |
| SogCLR (Chen & He, 2021) | 256 | ✗ | ✗ | 67.1 | SwAV Caron et al. (2020) | 4096 | ✗ | ✗ | **69.1** |
| SimCLR | 256 | ✗ | ✗ | 61.5 | SimCLR+ | 1024 | ✗ | ✗ | 68.0 |
| - w/ `Miracle` | 256 | ✗ | ✗ | 65.2 $_{+4.7}$ | - w/ `Miracle` | 1024 | ✗ | ✗ | 68.6 $_{+0.6}$ |

Table 9: Top-1 accuracy of `Miracle` with other SSL algorithms on CIFAR-10, CIFAR-100, STL-10, and ImageNet-100 with **linear probing**. All methods are pretrained for 200 epochs with batch size of 256. ResNet-18 (ResNet-50) is used as the backbone network for CIFAR/STL (ImageNet-100). ♥, ◇ denotes results taken from Wang *et al.* Wang et al. (2021), Zheng *et al.* Zheng et al. (2021), respectively. Cells, which report our implementation, are shaded in green. Our method achieves competitive results in all scenarios.

| Method | CIFAR-10 | CIFAR-100 | STL-10 | ImageNet-100 |
|---|---|---|---|---|
| SimCLR Chen et al. (2020a) ◇ | 84.9 | 59.3 | 85.5 | - |
| MoCo v2 He et al. (2020) ◇ | 86.2 | 59.5 | 85.9 | 76.6 ♥ |
| BYOL Grill et al. (2020) ◇ | 85.5 | 60.0 | 87.5 | 75.8 ♥ |
| SimSiam Chen & He (2021) ◇ | **88.5** | 57.8 | 87.5 | - |
| SimCLR | 82.5 | 56.4 | 83.8 | 70.6 |
| w/ `Miracle`-32 | 88.1 | **60.5** | **87.8** | **77.2** |

# D IMPLEMENTATION DETAILS

## D.1 SIMCLR+

ResNet-50 network is trained with LARS optimizer You et al. (2017) with weight decay of 0, momentum of 0.9. The learning rate is adjusted with square root scaling *i.e.,* the learning rate $\eta = 0.075 \times \sqrt{\text{BatchSize}}$. We use the batch size of 1024 which yield the learning rate of 2.4. On top of that, we adopt 4 layers MLP for projection head as suggested in Chen et al. (2021b). Specifically, each hidden layer is composed of linear layer (without bias) with the hidden dimension of 2048, batch normalization, ReLU activation. The projected dimension is 128.

**Linear evaluation.** For evaluation, we also use LARS optimizer with the same configuration. We train the linear classification with batch size of 1024 for 100 epochs. Note that for ImageNet-100, we observer that using normalized extracted features before the linear classifer yields slightly improvement for both SimCLR and `Miracle` (around 2%). Therefore, we leverage this setup for linear probing on ImageNet-100.

## D.2 OUR METHOD

**Projection head.** We adopt the projection head as in SimCLR Chen et al. (2020a) unless otherwise stated. Particularly, a Linear → ReLU → Batch Normalization → Linear → Batch Normalization. The hidden dimension is equal to input dimension *i.e.,* 2048 for ImageNet and 512 for CIFAR/STL.

**Data augmentations.** We adopt the data augmentation of Chen et al. (2020a) including: random cropping, horizontal flip, grayscale, color jittering, Gaussian blurring for pretraining. For linear probing, we simply use random cropping and horizontal flipping.

**MoCo setup.** We apply the same training recipe of SimCLR for MoCo in Table 1. In addition, we use the momentum $m = 0.99$ and queue size of 16384. Note that we use **symmetric** loss for training.

**Runtime performance.** Our method has similar training time to SimCLR. In our experiments, we do not observe any significant difference in timing compared to SimCLR during training.

# E  OTHER NUMERICAL RESULTS

The numerical results of the plots in the main paper are reported below. Particularly, we report the performance with different number of permutation $M$ in Table 10, the ablation study with learning rates in Table 11, the ablation study with different temperature values in Table 12, and the ablation study with different set sizes in Table 13.

Table 10: Comparing performance of `Miracle` with SimCLR and MoCo baselines on CIFAR-10, CIFAR-100, STL-10 with **k-nearest neighbors**. `Miracle`-n denotes our method when reordering/permuting the input $n$ times.

| | CIFAR-10 | | | | CIFAR-100 | | | | STL-10 | | | |
|---|---|---|---|---|---|---|---|---|---|---|---|---|
| Batch size | 128 | 256 | 512 | 1024 | 128 | 256 | 512 | 1024 | 128 | 256 | 512 | 1024 |
| SimCLR | 80.1 | 81.2 | 81.7 | 81.6 | 51.6 | 52.3 | 52.7 | 53.1 | 76.7 | 77.6 | 76.6 | 76.3 |
| Miracle-1 | 77.2 | 79.0 | 80.0 | 80.9 | 50.1 | 50.9 | 51.7 | 52.3 | 75.4 | 76.0 | 76.3 | 76.4 |
| Miracle-2 | 82.0 | 82.5 | 83.2 | 83.8 | 51.2 | 52.1 | 53.6 | 54.1 | 78.3 | 79.3 | 79.6 | 79.4 |
| Miracle-4 | 84.0 | 84.5 | 84.9 | 85.0 | 53.2 | 53.8 | 55.4 | 56.2 | 81.4 | 82.2 | 82.3 | 82.4 |
| Miracle-8 | 85.0 | 85.4 | 85.6 | 85.7 | 54.7 | 55.6 | 55.9 | 57.0 | 83.0 | 83.5 | 83.6 | 83.6 |
| Miracle-16 | 85.3 | 86.0 | 86.0 | 85.8 | 55.3 | 56.2 | 56.5 | **57.1** | 84.2 | 84.4 | 84.2 | **84.7** |
| Miracle-32 | **85.8** | **86.2** | **86.2** | 86.1 | **55.4** | **56.6** | **57.1** | **57.1** | **84.7** | **85.2** | **84.6** | 84.5 |

Table 11: Performance of `Miracle` and SimCLR on CIFAR-10/CIFAR-100 and STL-10 with **k-nearest neighbors** when varying the learning rate. Batch size of all experiments are set to 256 and the number of pretraining epoch is 200.

| | CIFAR-10 | | | | CIFAR-100 | | | | STL-10 | | | |
|---|---|---|---|---|---|---|---|---|---|---|---|---|
| Learning rate | 0.03 | 0.06 | 0.09 | 0.12 | 0.03 | 0.06 | 0.09 | 0.12 | 0.03 | 0.06 | 0.09 | 0.12 |
| SimCLR | 81.2 | 81.7 | 81.1 | 80.7 | 52.3 | 52.2 | 51.5 | 50.9 | 77.6 | 77.8 | 77.2 | 77.2 |
| Miracle-32 | **85.8** | **86.4** | **87.0** | **86.9** | **56.6** | **57.4** | **57.4** | **57.4** | **85.2** | **84.5** | **84.5** | **84.6** |

Table 12: Performance of `Miracle` and SimCLR on CIFAR-10/CIFAR-100 with **k-nearest neighbors** when varying the temperature. Batch size of all experiments are set to 256. `Miracle`-n denotes our method when reordering/permuting the input $n$ times.

| | CIFAR-10 | | | | | | CIFAR-100 | | | | | |
|---|---|---|---|---|---|---|---|---|---|---|---|---|
| Temperature | 0.03 | 0.05 | 0.07 | 0.1 | 0.2 | 0.5 | 0.03 | 0.05 | 0.07 | 0.1 | 0.2 | 0.5 |
| SimCLR | 79.6 | 79.2 | 81.2 | 83.2 | 85.7 | 84.2 | 52.4 | 51.4 | 52.3 | 52.7 | 52.4 | 48.4 |
| Miracle-32 | **86.4** | **85.9** | **86.2** | **86.6** | **86.5** | **84.1** | **56.5** | **57.5** | **56.6** | **55.5** | **53.1** | **47.5** |

Table 13: Performance of `Miracle` with **k-nearest neighbors** evaluation compared to SimCLR when varying number of elements per set. All models are pretrained with batch size of 256 for 200 epochs. The accuracy of SimCLR on CIFAR-10 and CIFAR-100 is 81.2 and 52.3, respectively.

| | CIFAR-10 | | | | | CIFAR-100 | | | | |
|---|---|---|---|---|---|---|---|---|---|---|
| Set size | 2 | 3 | 4 | 5 | 6 | 2 | 3 | 4 | 5 | 6 |
| w/ Miracle-1 | 79.0 | 76.9 | 75.1 | 72.3 | 71.1 | 50.1 | 43.4 | 41.0 | 38.4 | 36.4 |
| w/ Miracle-32 | 86.2 | 86.3 | 86.3 | 86.0 | 85.9 | 56.6 | 56.2 | 55.9 | 55.7 | 55.0 |

