# OpenReview forum: "Set Discrimination Contrastive Learning"
_ICLR.cc/2023/Conference — Submitted to ICLR 2023_

### Official Review · Reviewer_1LDS · 2022-10-24

**Confidence:** 3
**Correctness:** 2
**Technical Novelty And Significance:** 3
**Empirical Novelty And Significance:** 2
**Recommendation:** 5

**Clarity, Quality, Novelty And Reproducibility:**

The paper is well-written and has nice clarity. The work has nice originality. It is interesting to see that set-based representation learning is more effective than instance-level representation learning in the context of contrastive learning.

However, the underlying mechanism of why it works remains unknown. The author tries to provide some explanation through theoretical analysis. But I am afraid that the current theoretical analysis is superficial and not satisfying. Together with the issue of the experiment section mentioned above, I find the paper's quality needs to be improved.

**Strength And Weaknesses:**

Pros:

1. The paper is easy to follow. The paper has nice figures and beautiful tables.

2. Author provides a nice ablation of the proposed method.

Cons:

1. Theoretical interpretation is vague.
- 1(a) Theorem 1: as shown in the proof, theorem 1 basically says "InfoNCE(Z,Z') <= I(Z; (X1,X2))". The theorem is very general and not specific to the proposed algorithm. For example, the theorem holds no matter whether data augmentation is used or not. However, data augmentation in CL contributes a lot to the success of CL
- 1(b) The author use theorem 1 to argue that "If the encoder learns features that are specific to only one instance of the set, it does not increase the mutual information with any of the other instances of that set." Such an argument is not precise and clear. First, there is no such thing called "mutual information with instance". Mutual information is defined between random variables, not instances. Second, by symmetry, it is easy to see that I(Z; X1) = I(Z; X2) since X1 and X2 are two independent random samples from data. Thus there is no such case where I(Z; X1) is high while I(Z; X2) is low.
- 1(c) hard negatives mining: The logic in the paragraph "Set discrimination from hard negatives perspective." does not make sense to me. I do not see the definition of "hard negative" and do not see why "set construction" is a strategy to produce hard negatives.

2. Questions on experiments
- 2(a) Uncommon evaluation metrics: Most of the contrastive learning paper report "linear probing" results, e.g., the baselines used in this work, SimCLR, MoCoV2, DCL, FlatNCE. I suggest the author use linear probing instead of KNN as their main evaluation metric for CIFAR-10/-100, and STL-10 experiments.
- 2(b) Number inconsistency from prior works: I find the baseline results reported by the author are not consistent with that reported in the original paper. In this paper's table 1, CIFAR-10, batchsize=256, the KNN accuracy of DCL is 82.7. However, in the original DCL[1] paper, the accuracy is 84.2 (Table 2).

[1] Yeh, Chun-Hsiao, et al. "Decoupled contrastive learning." arXiv preprint arXiv:2110.06848 (2021).

**Summary Of The Paper:**

The paper proposes a new way to do contrastive learning (CL). Unlike prior CL methods that focus on instance-level representation, this work contrast sets of representation. Experiments of CIFAR-10/-100 and Imagenet-100 show the proposed method's effectiveness.

**Summary Of The Review:**

Overall, considering the paper's originality and quality, I think the current version of the paper does not reach the bar of being published in ICLR. However, I will reconsider my score if all my concerns get addressed.

---

> ### Author Response · Authors · 2022-11-18
> **Thanks for your review. Here are our answer**
>
> ### 1. The theorem is very general and not specific to the proposed algorithm. For example, the theorem holds no matter whether data augmentation is used or not. However, data augmentation in CL contributes a lot to the success of CL
>
> Theorem 1 is built upon the concept of sets and instances, which is specific to our set-based contrastive learning.  In our explanation, MIRACLE encourages common features and avoids low-level image features. On the other hand, strong data augmentation enforces the network to learn (augmentation) invariance features. Thus, they are complementary and orthogonal to each other. As can be observed from Table 4, albeit SimCLR and MIRACLE both suffer from weaker augmentation but MIRACLE does offer advantages over the baseline.
>
> ### 2. First, there is no such thing called "mutual information with instance". Mutual information is defined between random variables, not instances. Second, by symmetry, it is easy to see that I(Z; X1) = I(Z; X2) since X1 and X2 are two independent random samples from data. Thus there is no such case where I(Z; X1) is high while I(Z; X2) is low.
>
> Thanks for spotting the first point, we have amended this in the revised version.
>
> Regarding the second point, we would like to clarify that by "the global context is encouraged to have high mutual information with all the instances of that set", we are referring to the term $I(X_1; X_2 | Z)$ i.e., additional information that image $X_1$ captures about $X_2$ when we already had the set representation. The feature $Z$ has to carry the information for a *pair* of $X_1, X_2$ in order to maximize this conditional mutual information since two images are independent of each other.
>
> ### 3. The logic in the paragraph "Set discrimination from hard negatives perspective." does not make sense to me. I do not see the definition of "hard negative" and do not see why "set construction" is a strategy to produce hard negatives.
>
> We use the definition in [1] for *hard negative* samples. Specifically, they are samples "that are difficult to distinguish from an anchor point".
>
> An example of hard negative samples is to construct different sets from the same group of instances, as explained in Section 3.2 in the main paper. In short, these sets are similar (thus, hard) because they are composed of the same instance paired with another ones (i.e., partially overlap).
>
> ### 4. Uncommon evaluation metrics: Most of the contrastive learning paper report "linear probing" results, e.g., the baselines used in this work, SimCLR, MoCoV2, DCL, FlatNCE. I suggest the author use linear probing instead of KNN as their main evaluation metric for CIFAR-10/-100, and STL-10 experiments.
> Thanks for your comment. In our observation, many prior works that conducted experiments *for small-scale datasets* such as [2, 3, 4, 5] adopt this metric, so we simply follow them. Nevertheless, we also report the performance of MIRACLE (and baseline) with linear probing in Table 9. We will consider moving this table to the main paper.
>
> ### 5. Number inconsistency from prior works: I find the baseline results reported by the author are not consistent with that reported in the original paper. In this paper's table 1, CIFAR-10, batchsize=256, the KNN accuracy of DCL is 82.7. However, in the original DCL[1] paper, the accuracy is 84.2 (Table 2).
>
> Since we could not find the official implementation of the DCL paper, we resort to the public repo [here](https://github.com/raminnakhli/Decoupled-Contrastive-Learning) to reproduce the results (which we have verified the correctness). Nonetheless, we believe our reproduced results are still interesting because 1) the performance of DCL + MIRACLE outperforms the original paper (e.g., 86.6 vs 84.2 in this example), and 2) these experiments validate the effectiveness of MIRACLE when applied with different variants of InfoNCE.
>
>
>
> **References**:
>
> [1] Contrastive Learning with Hard Negative Samples (ICLR 2021)
>
> [2] Unsupervised Feature Learning via Non-Parametric Instance Discrimination (CVPR 2018)
>
> [3] Unsupervised Embedding Learning via Invariant and Spreading Instance Feature (CVPR 2019)
>
> [4] Unsupervised Feature Learning by Cross-Level Instance-Group Discrimination (CVPR 2021)
>
> [5] Decoupled Contrastive Learning (ECCV 2022)

---

> > ### Comment · Reviewer_1LDS · 2022-12-05
> > **Thanks for your response**
> >
> > I appreciate the rebuttal provided by the author. However, my questions are not well addressed. Thus, I am afraid I would recommend not accepting this work.
> >
> > For example, in my original review, I explained why the sentence is not precise and unclear, "If the encoder learns features that are specific to only one instance of the set, it does not increase the mutual information with any of the other instances of that set." The author claimed they amended this issue in the revision. However, the sentence is untouched in the revision.

---

### Official Review · Reviewer_LKL2 · 2022-10-25

**Confidence:** 4
**Correctness:** 3
**Technical Novelty And Significance:** 2
**Empirical Novelty And Significance:** 3
**Recommendation:** 5

**Clarity, Quality, Novelty And Reproducibility:**

Clear, and well-written. Though the idea comes from previous work such as the pooling model, it seems not sufficiently novel in theory, but it improves the performance in practice. The authors didn’t mention code releasing, however, the Algorithm is well explained.

**Strength And Weaknesses:**

The paper is well-written and easy to follow. The review is clear and the discussion between the existing work and this work is provided.

One question I have is what’s the difference between permutating $B$ batch $M$ times and letting the model process your data $B*M$ times? If a baseline model runs $M$ times the number of epochs, would the performance increase? I am not convinced about the experimental results by comparing all methods for 200 epochs since this method feeds the data M times compared with other methods.

Another point is that the theory behind the work is limited. Theorem 1 in the paper does not provide enough insight to explain the proposed method (basically a standard way to show the upper bound, but no explanation of the usage).

**Summary Of The Paper:**

Starting from the limitation that unnecessary information could be introduced into the pretext task learning in contrastive learning, this work proposed a method to learn shared features by maximizing set mutual information and circumventing instance discrimination. The set pooling model $g(\phi( ))$ process multiple output of encoder. This function itself has been proved to have permutation invariant properties. This work generates a set of data first and applies the aggregation function to a set of outputs. Since randomly grouping data in a mini-batch to a set will reduce the number of positive set pairs and negative set pairs, the work proposed to permute input multiple times to increase the number of set pairs.

**Summary Of The Review:**

The lack of theory and the unclear comparison makes me not quite sure about the real contribution of the paper. This is a good application of some existing methods though.

---

> ### Author Response · Authors · 2022-11-18
> **Thanks for your review. Here are our answer**
>
> ### 1. One question I have is what’s the difference between permutating $B$ batch $M$ times and letting the model process your data $B ∗ M$ times? If a baseline model runs $M$ times the number of epochs, would the performance increase? I am not convinced about the experimental results by comparing all methods for 200 epochs since this method feeds the data M times compared with other methods.
>
> Our permutation is only applied to **extracted** feature and group them to construct a set. Therefore compare to processing the data $B * M$ times our approach does not lead to $M$ times more computation for extracting features nor more parameter updates. Please see our response to question 1 in response to common questions.
>
> ### 2. Another point is that the theory behind the work is limited. Theorem 1 in the paper does not provide enough insight to explain the proposed method (basically a standard way to show the upper bound, but no explanation of the usage).
>
> We'll add more clarification about our interpretation in Section 3.4 as in our response to Reviewer 1LDLS. With this intuitive explanation, empirical results, and the simplicity of our approach, we believe this method is useful to the community and could inspire more research for more rigorous theoretical results, similar to recent findings to explain SimSiam in [1].
>
> **References**:
>
> [1] How Does SimSiam Avoid Collapse Without Negative Samples? A Unified Understanding with Self-supervised Contrastive Learning (ICLR 2022)

---

### Official Review · Reviewer_VebJ · 2022-10-25

**Confidence:** 3
**Correctness:** 3
**Technical Novelty And Significance:** 3
**Empirical Novelty And Significance:** 3
**Recommendation:** 5

**Clarity, Quality, Novelty And Reproducibility:**

The paper is clearly written. The proposal of set contrastive learning is novel and new to me. But I do not see the significance of the proposal on the image classification task, since the permutation invariance is kind of counter intuitive to me. Also see above weakness in terms of this issue.

**Strength And Weaknesses:**

Strength:

The paper is well written. The paper proposes to impose the permutation invariance as constraint in training self-supervised learning methods. Inspired by set-based feature learning, the paper propose to aggregate set features from individual sample features by a symmetric function. Empirical results demonstrates superiority of the proposed method.


Weakness:

W1: M is the number of times the approach shuffle the input. This incurs an issue of fairness. That being said, the proposed miracle method is exposed to the training data which is M-1 times (larger batchsize) more than its baselines. This empirical comparison seems unfair to the baselines. What happened to the conventional methods such as SimCLR if they also see the same amount of training data (with the same batchsize), i.e., also are trained with those additional permutated data (but without permutation invariance)?

W2: I am also confused why permutation invariance is necessary on traditional image classification task, too. Unlike 3D point applications,  images are independent to each other, I am not convinced why keeping set permutation invariance for images in a set manner should be beneficial for image classification task, since classification prediction is independently deployed on each individual test images. This is counter intuitive to me. Please kindly address this issue during the rebuttal.


**Summary Of The Paper:**

This paper proposes to introduce the set permutation invariance constraint in training self-supervised learning approaches. The paper also investigates the effect of using different aggregation function that encourages the permutation invariance among images. Empirical results demonstrates certain benefits of the proposed scheme in comparison to the SOTA SSL methods.

**Summary Of The Review:**

Owing to the fairness issue, I am afraid I cannot recommend acceptance at this time. I might increase my score after rebuttal, if the fairness issue and the intuition behind the set invariance constraint can be properly addressed.

---

> ### Author Response · Authors · 2022-11-18
> **Thanks for your review. Here are our answers**
>
> ### 1. $M$ is the number of times the approach shuffle the input. This incurs an issue of fairness. That being said, the proposed miracle method is exposed to the training data which is $M-1$ times (larger batch size) more than its baselines. This empirical comparison seems unfair to the baseline
> We would like to clarify that Miracle **does not** have more exposure to training data than the baseline. Please see our response to the common question.
>
>
> ### 2. I am also confused why permutation invariance is necessary on traditional image classification task, too.
>
> We would like to point out that our approach is not meant to enforce the permutation invariance for image classification. Precisely, our goal is to use set-structured learning to encourage learning common features across instances in contrastive learning, which we hypothesize to be useful for different downstream tasks.  In this paper, we use image classification as a downstream task to demonstrate the effectiveness of our set-based learning. Since every element of a set need to be permutation invariance by definition (and having an order when group images does not make intuitive sense), we simply adopt the standard (permutation invariance) aggregation approach in set-structure learning.

---

> > ### Comment · Reviewer_VebJ · 2022-12-05
> > **Thanks for your response.**
> >
> > I sincerely appreciate the response from the answers. However, I am still concerned with the motivation of the paper, i.e., why the set permutation task would improve the generalization of features. In this regard, I am afraid I would maintain my score.

---

### Official Review · Reviewer_g2Gv · 2022-10-25

**Confidence:** 4
**Correctness:** 3
**Technical Novelty And Significance:** 3
**Empirical Novelty And Significance:** 2
**Recommendation:** 5

**Clarity, Quality, Novelty And Reproducibility:**

**Clarity/Quality**

- The paper is easy to follow, and contains a lot of numerical results, albeit on small scale datasets. A lot of crucial results to judge the impact/limitations of the method are however hidden in the supplement, e.g. the comparison to other methods on ImageNet-1k where set contrastive learning falls short.

**Novelty**

- There might be similar methods that aggregate feature representations prior to contrastive learning. Especially the relationship to prototype-based methods (Swav, Moco) could be better discussed, as e.g. a momentum encoder probably has a similar effect on the feature representations.
- Besides this, I believe that the loss can be considered "novel", although in the used form with only two representations it looks more like an incremental improvement to SimCLR.

**Reproducibility**

- The authors do not state if a reference implementation will be publicly available.
- From the description in the paper, it is probably possible to re-implement the method.

**Strength And Weaknesses:**


**Major Weaknesses**

- The proposed framework (and claims) are quite general, while the actual experiments only test a very specific subset of "set contrastive learning": The number of features to aggregate is set to $K = 2$, and it is unclear if larger sets actually improve or degrate the performance (Figure 3). I wonder if having a "set" is the crucial detail here, or whether "pairs" of representations are sufficient.
  - In addition, I am unsure if the very general paper title "Set discrimination contrastive learning" is justified. Right now, the authors mainly show improvements when using image pairs rather than sets.
- The experimental validation is quite sparse. The paper is mostly evaluated on small scale data. A full ImageNet experiment with comparisons to multiple baselines is missing. Table 3 should be extended with more self-supervised learning algorithms, including non-contrastive ones, like mentioned in the appendix.
- MoCo is mentioned in the conclusions, but only included in Table 1. Either this claim should be dropped, or Tables 2 and 3 should include MoCo as a comparison method as well. In general, Table 8 should be included in the main paper as it shows the important limitations that other self-supervised learning schemes outperformed set contrastive learning. Also, some methods, e.g. DINO (Caron et al., 2021) are missing which demonstrate accuracies of up to 75.3%. Similarly, training MoCo-v2 longer alo gives another boost to 71.1%, which is also hidden in this table.
- The conclusion mentions increased "robustness": Where is this demonstrated in an experiment?

**Minor Weaknesses**

- The algorithm 1 should be better typeset. Right now there is a lot of confusion between math and pseudocode notation. Either is fine, as long as the syntax is consistent. It might be useful to add comments to the pseudo-code.
- There is no statement on hyperparameter selection.
- The bar plots are highly misleading as the y axis does not start at zero. Figure 3, 4 can be better represented by a lineplot (or a table), ideally with error bars to check which differences are meaningful.
- Even though small scale experiments are used, no error bars / standard deviations are reported.
- The authors claim "signficant" improvements at least in one location in the paper, but do not perform a suitable statistical test to back up this claim. The wording should be adapted (e.g., "considerable") or a test needs to be performed.

**Additional Questions**

- Did you ran your experiments using $K = 1$ to verify the baseline implementation? This setup should exactly match the baseline, correct?
- A number of works learn prototypes for contrastive learning, which effectively summarize multiple features into a single representation. Could you outline why set contrastive learning would yield improvements over such methods? For example, in Table 3 (ImageNet-1K), MoCo-V2 would outperform Miracle (68.6 vs. 71.1).
- Is it possible that the method increases the effective batch size by a factor of 2? This is a sensitive parameter in e.g. SimCLR, I wonder whether this might be a confounder for the experimental results.
- Is there a potential advantage to the method besides a boost in classification performance that I should be aware of and have not considered in my review?
- Did you start from a particular reference implementation of SimCLR? If so, could you state this in the paper/supplement to facilitate reproducibility?



**Summary Of The Paper:**

The authors propose a contrastive learning scheme where not the feature representations of individual samples, but an aggregated feature representation of multiple samples (a set) is considered. The authors claim improvements over a SimCLR baseline on CIFAR, STL, ImageNet-100 and full ImageNet.

**Summary Of The Review:**


The method proposed by the paper is interesting, but the execution and empirical validation is insufficient. Specifically, only a very particular (and not very interesting) special case of the set contrastive learning setup is explored, and mostly validated on small scale datasets. The method seems to be effective and improves the baseline, but falls short in comparison with reference results from the literature. The authors should focus to better position their work within the self-supervised learning literature, and also include more commonly used contrastive and non-contrastive models to compare to in Table 7. There is a lot of follow up work on SimCLR (which is the baseline here) and it is very likely that other "additions" to SimCLR have similar effect to the set-contrastive learning scheme proposed here.

Since the learning approach is overall intersting, I am willing to adapt my score based on the discussion with authors and other reviewers if an interesting application/advantage of the method beyond SOTA results can be demonstrated.

---

> ### Author Response · Authors · 2022-11-18
> **Thanks for your review. Here are our answers**
>
> ### 1. The number of features to aggregate is set to K=2, and it is unclear if larger sets actually improve or degrade the performance (Figure 3). I wonder if having a "set" is the crucial detail here, or whether "pairs" of representations are sufficient. In addition, I am unsure if the very general paper title "Set discrimination contrastive learning" is justified. Right now, the authors mainly show improvements when using image pairs rather than sets.
>
> In Figure 3 of the main paper, it can be seen that all K values improve upon the SimCLR baseline. Therefore, we believe that having a set is beneficial.  However, in our experiments, a set is constructed by randomly selecting samples of a mini-batch. Therefore, a large set of samples does not necessarily encourage more common features because of their different semantics, thus, K=3 case is just as good as K=2. As future work, we can incorporate clustering/nearest-neighbors search into training or some other techniques, increasing might be more beneficial.
>
> ### 2. A full ImageNet experiment with comparisons to multiple baselines is missing. Table 3 should be extended with more self-supervised learning algorithms. In general, Table 8 should be included in the main paper as it shows the important limitations that other self-supervised learning schemes outperformed set contrastive learning. Also, some methods, e.g. DINO (Caron et al., 2021) are missing which demonstrate accuracies of up to 75.3%. Similarly, training MoCo-v2 longer alo gives another boost to 71.1%, which is also hidden in this table.
>
> Thanks for your suggestion. We would like to point out that in our experiments, we set the training epoch on ImageNet to 200 epochs whereas DINO (and most approaches that have top-1 accuracy over $70\%$) has much longer training. Thus, we only report performance of methods that shares the same budget. That being said, we will move the Table 8 to the main paper.
>
> ### 3. The conclusion mentions increased "robustness": Where is this demonstrated in an experiment?
>
> By robustness, we mean the consistency in the improvement over the baseline approach across datasets and hyper-parameters setup. We will amend this wording.
>
> ### 4. There is no statement on hyperparameter selection.
> For all hyper-parameters aside $K$ & $M$, we adopt from previous works such as [3, 4]. We run a grid search on CIFAR-10 in the range of $[2, 4, 8]$ and $[2, 4, 8, 16, 32]$ for $K$ and $M$, respectively. Then, we use the same configurations across datasets.
>
> ### 5. The bar plots are highly misleading as the y axis does not start at zero. Figure 3, 4 can be better represented by a lineplot (or a table), ideally with error bars to check which differences are meaningful. Even though small scale experiments are used, no error bars / standard deviations are reported. The authors claim "signficant" improvements at least in one location in the paper, but do not perform a suitable statistical test to back up this claim. The wording should be adapted (e.g., "considerable") or a test needs to be performed.
>
> Thanks for your suggestion, we will amend Figure 3 and 4 accordingly and update the confidence interval (CI) in the appendix. The 95-CI is roughly $0.3$ for all experiments (we noted this in the caption of Table 1 because of the lack of space), thus, we believe the improvement is significant. The CI is calculated with 3 runs.
>
> ### 6. Did you ran your experiments using $K=1$ to verify the baseline implementation? This setup should exactly match the baseline, correct?
>
> Yes, we did and the results match SimCLR baseline. Please see our response for question 1 in common questions.
>
>
> ### 7. A number of works learn prototypes for contrastive learning, which effectively summarize multiple features into a single representation. Could you outline why set contrastive learning would yield improvements over such methods? For example, in Table 3 (ImageNet-1K), MoCo-V2 would outperform Miracle (68.6 vs. 71.1).
>
> First, we would like to point out that MoCo-V2 training for $800$ epochs to get the performance of $71.1$ where we only train the model for $200$ epochs.
>
> Regarding the performance, we would like to point out some advantages of our set discrimination compared to clustering below.
>
> First, Miracle could also be viewed as a mechanism for augmentation (as in Section 3.4). Moreover, there are recent works demonstrate the effectiveness of mixing data for self-supervised/contrastive learning [1, 2]. These approaches mostly work on the image space while our approach operates on latent space and can be agnostic to different modalities.
>
> Second, the set construction offers a clean formulation to augment the number of positive/negative samples in a batch without incurring much overhead (because every grouped set is a different data point). On the other hand, previous methods rely on a momentum memory or increasing batch size to achieve this goal.

---

> > ### Author Response · Authors · 2022-11-18
> > **Additional answers**
> >
> > ### 8. Is it possible that the method increases the effective batch size by a factor of 2? This is a sensitive parameter in e.g. SimCLR, I wonder whether this might be a confounder for the experimental results.
> > We use the same batch size (as well as other hyperparameters) as SimCLR and only modify the code to process extracted feature, please see our response to common questions for detailed implementation.
> >
> > ### 9. The method seems to be effective and improves the baseline, but falls short in comparison with reference results from the literature. Is there a potential advantage to the method besides a boost in classification performance that I should be aware of and have not considered in my review?
> > Please see our response 7. In addition, we believe that MIRACLE is useful in the following aspects. Our set discrimination task is a simple, effective, easy-to-implement method to improve the performance of contrastive learning. Our method is orthogonal to the SoTA methods on ImageNet-1k and so the performance could potentially be even better in future research. For example, one can explore our method for ViT network e.g., by constructing sets from patches of different samples via mixing the attention input. In a long run, we believe that our method is sustainable and could inspire more future works.
> >
> > ### 10. The authors do not state if a reference implementation will be publicly available. Did you start from a particular reference implementation of SimCLR? If so, could you state this in the paper/supplement to facilitate reproducibility?
> >
> > We will release the full implementation upon acceptance. Our implementation is based on [this SimCLR repository](https://github.com/leftthomas/SimCLR) and modifies: 1) setting the optimizer to SGD instead of Adam and temperature to $0.07$ to match the setup in our experiments 2) implementing Miracle according to the PyTorch code in our response to common questions. For data augmentation, we adopt from [CLD](https://github.com/frank-xwang/CLD-UnsupervisedLearning) for precise reproduction.
> >
> >
> >
> >
> > **References**:
> >
> > [1] TokenMix: Rethinking Image Mixing for Data Augmentation in Vision Transformers (ECCV 2022)
> >
> > [2] RegionCL: Exploring Contrastive Region Pairs for Self-supervised Representation Learning (ECCV 2022)
> >
> > [3] Unsupervised Feature Learning by Cross-Level Instance-Group Discrimination (CVPR 2021)
> >
> > [4] Decoupled Contrastive Learning (ECCV 2022)

---

### Author Response · Authors · 2022-11-18
**Response to common question**

We would like to thank all reviewers for spending time evaluating our work and providing constructive feedback. We appreciate that the reviewers found our work novel (Reviewer g2GV, VebJ) and interesting. All reviewers unanimously agree that the paper is polished and easy to understand. Let us now first address the common concerns raised by the reviewers and then respond to the questions from individual reviewers, below.

### Unfair comparison because MIRACLE exposes to the data more than SimCLR

We would like to clarify that the proposed approach does **not** expose more training data/have a higher number of training epochs than the baseline. Particularly, we only "permute" the extracted features of images from a batch to construct different sets from the **same** batch of images. For clarity we included our implementation of MIRACLE and the baseline SimCLR in PyTorch below:

```
 # feature extracting
 views_1, views_2 = batch
 feature_1, projected_1 = ResNet(views_1) # projected_1 = projection_head(feature_1)
 feature_2, projected_2 = ResNet(views_2)

 # SimCLR
 projected_1, projected_2 = F.normalize(projected_1), F.normalize(projected_2)
 loss = InfoNCE(projected_1, projected_2)

 # MIRACLE
 # randomly init a permutation matrix
 perm_mat = torch.argsort(torch.rand((M, B)), dim=-1).reshape(-1) # M: number of permutation, B: batch size
 out_1 = out_1[perm_mat].reshape(B * M//K, K, D) # K: number of elements per set, D: projection dim
 out_2 = out_2[perm_mat].reshape(B * M//K, K, D)
 # pooling, we can replace mean function with any other aggregation functions as in Tab. 5
 out_1, out_2 = out_1.mean(dim=1), out_2.mean(dim=1)
 out_1, out_2 = F.normalize(out_1, dim=-1), F.normalize(out_2, dim=-1)
 loss = InfoNCE(projected_1, projected_2)

 # gradient descent
 loss.backward()
 optimizer.step()
```


We could do the same for conventional SimCLR (i.e., setting K=1 as reviewer g2GV has mentioned) but that would not bring any benefits because all losses of *repeated* instances are equal, thus, the loss is simply multiplied by a constant $M$. Therefore, the performance in this case will be similar to SimCLR.
More formally, the objective of SimCLR is given by:

$$L_{SimCLR} = - \frac{1}{2B}\sum_i^{2B}\log\frac{\exp(x_i^Tx_{i'})}{\exp(x_i^Tx_{i'})+\sum_{j\neq i}\exp(x_i^Tx_j)}$$


When setting the value of $K$ to $1$ the loss function is reduced to:
$$L = - \frac{1}{2\cdot M\cdot B}\sum_i^{2B}    M\log\frac{\exp(x_i^Tx_{i'})}{M\big[\exp(x_i^Tx_{i'})+\sum_{j\neq i}\exp(x_i^Tx_j)\big]}$$

In contrast, when we permute, group, and aggregate these instances to create a set, every permutation constitutes different set, thus, there are differences in the gradient for each repeated instance w.r.t InfoNCE loss for a set.

In addition, in terms of efficiency, MIRACLE only incurs a small overhead because of: the set aggregation operators (e.g., sum, mean, etc) and additional computation to calculate InfoNCE loss.

---

### Decision · Program_Chairs · 2023-01-20

**Decision:**

Reject

**Justification For Why Not Higher Score:**

The paper needs to undergo another round of major revision - the current sets of analyses and experimental results do not explain how the method works and are insufficient for verifying whether the proposed method is effective or not.

**Justification For Why Not Lower Score:**

N/A

**Metareview: Summary, Strengths And Weaknesses:**

This paper proposes a contrastive self-supervised learning framework based on the set encoding of the positive and negative. Specifically, the proposed framework performs contrastive learning using the set encodings, where the positive sets are constructed by applying different augmentations to the same set of instances, while the negative sets are constructed from different instances. The authors theoretically show that set-discrimination prioritize common features across instances over low-level features from each instance, and experimentally validate the proposed method method on multiple benchmark datasets, against multiple contrastive learning baselines, which it outperforms.

The reviewers considered the idea of introducing the set encoding to self-supervised learning as novel, the idea of generating different permutations and enforcing permutation invariance to the different set encodings as reasonable, and the paper well-written.

However, at the same time, the reviewers had concerns about weak experimental validation, as the experiments were done mostly with small set size (K=2), on small-scale datasets, against only a few baselines, missing out most state-of-the-art self-supervised learning methods. The reviewers were also concerned about the lack of analysis on how the proposed set permutation helps with the generalization. There were also concerns regarding unfair comparison as the reviewers misunderstood that the set permutation requires more computations and exposes the model to a larger amount of data.

During the discussion period the misunderstandings on the fairness of the experimental validation were cleared away. However, the reviewers kept their unanimous weak reject ratings even after the discussion period, since they had remaining concerns on unclear motivation as to how the set-encoding helps with generalization, and weak experimental validation. The authors are advised to focus on addressing these two critical weaknesses in their revision.